# Biodegradable Conductive Layers Based on a Biopolymer Polyhydroxybutyrate/Polyhydroxyvalerate and Graphene Nanoplatelets Deposited by Spray-Coating Technique

Sandra Lepak-Kuc [1,2,*], Katarzyna Wójkowska [1], Dorota Biernacka [1], Aleksandra Kądziela [1,2], Tomasz Tadeusz Murawski [1], Daniel Janczak [1,2] and Małgorzata Jakubowska [1,2]

[1] Faculty of Mechanical and Industrial Engineering, Warsaw University of Technology, Narbutta 85, 02-524 Warsaw, Poland; katarzyna.wojkowska.stud@pw.edu.pl (K.W.); dorota.biernacka.stud@pw.edu.pl (D.B.); aleksandra.kadziela.dokt@pw.edu.pl (A.K.); tomasz.murawski.stud@pw.edu.pl (T.T.M.); daniel.janczak@pw.edu.pl (D.J.); malgorzata.jakubowska@pw.edu.pl (M.J.)

[2] Centre for Advanced Materials and Technologies (CEZAMAT), Warsaw University of Technology, 02-822 Warsaw, Poland

[*] Correspondence: sandra.kuc@pw.edu.pl

**Abstract:** In light of the growing concern for environmental protection and the alarming amount of waste produced due to hygiene regulations, this study suggests a biodegradable and eco-friendly solution that could make a significant contribution to the preservation of our planet. The developed solution was based on a polyhydroxybutyrate/polyhydroxyvalerate biopolymer, which has been tested regarding its physicochemical parameters and possible use in printed electrically conductive structures. Graphene nanoplatelets have been used as the conductive functional phase, due to literature reports of their potential use in biomedical applications and due to the potential of providing cytocompatibility in electrical structures by carbon nanomaterials. Prepared composites have been spray-coated onto PET film and paper substrates and then subjected to electrical, adhesion and optical measurements. In order to establish the conductivity of the developed composite, its resistance, layer thickness and surface topography were measured. Optical parameters have been specified using scanning electron microscopy (SEM) imaging and spectrophotometry. The conducted research opens a wide path for the use of the polyhydroxybutyrate/polyhydroxyvalerate biopolymer with graphene nanoplatelets in biomedical applications, ensuring good conductivity, biocompatibility and stability.

**Keywords:** biopolymer; graphene nanoplatelets; electrical properties; green electronics

## 1. Introduction

Increasing social awareness of healthcare and environmental problems provides scientists with the opportunity to develop biodegradable and biocompatible biomedical devices. Given the issue of an ageing population and the growing resistance of viruses to existing drugs, it becomes important to develop implantable electronic devices with the task of precisely targeting the human body. Establishing new technologies for smart and flexible sensing systems will provide the healthcare industry with easier and more effective methods to diagnose and take care of patients. It may provide a completely remote and safe way of monitoring people's vital signs [1]. There are studies on applications for almost all body parts. For example, there are devices for deep brain stimulation that target neurons with minimal damage to other brain regions, ultrathin needle-like eye implants to replace damaged retinas, cochlear implants to restore patients' hearing, cardiac pacemakers for postoperative control of cardiac rate and rhythms that undergo complete dissolution and clearance by natural biological processes after a defined operating timeframe, devices for nerve stimulation in the spinal cord, the Utah electrode array for monitoring and treatment of neurological disorders and many other implantable devices for sensing and monitoring

blood parameters such as pulse, alcohol concentration or oxidation [2–5]. The use of this kind of implantable nano-device may reduce many risks associated with currently used surgical methods and possible postoperative infections.

Nanomaterials will play the main lead in revolutionizing the technology of manufacturing electronic biomedical devices. By now they are successfully becoming a part of many industries related to everyday objects, such as food safety, environmental sciences, cosmetics and, most importantly, healthcare. Nanomaterials are widely researched due to their unique biological and physicochemical properties emphasizing their biocompatibility, stability and very low cytotoxicity [6]. Properties such as biocompatibility and cytotoxicity are dependent on aspects such as the size of particles, composition, functionalisation degree or structure, so it is very important to examine every nanomaterial's ability to interact with cells, tissues or a living body [7]. One of the major nano-sized conductive materials adaptable to printed electronics technologies are graphene nanoplatelets [8,9]. Many sources name graphene as the 'miracle material' due to its remarkable properties [10–12]. Graphene nanoplatelets combine large-scale production and low costs with remarkable physical properties. They are composed of single and few-layer graphene mixed with thicker graphite; hence, structurally they can be classified between graphene and graphite. The thickness of graphene nanoplatelets can vary from 0.34 up to 100 nm and they can be easily included in polymeric matrices, which makes them perfect for use in the field of printed electronics [9,13–15]. Recently, it has been found that the use of graphene nanoplatelets as a tissue-engineering scaffold promotes the attachment and proliferation of some cell lines, and they also show antibacterial activity against a wide range of bacteria [12]. Considering biomedical applications, there is a problem with the hydrophobic nature of graphene, which makes it impossible to interact safely with human cells and tissues. The use of a biocompatible matrix makes it highly stable in water and ensures good cytocompatibility and biodegradability. Here, a promising future opens up for the use of biopolymers [7].

Biopolymers are organic substances derived from living organisms or biological resources, for example, plants, animals or microorganisms. They are renewable, environmentally friendly and biodegradable, but their tensile strength, impact strength and thermal stability are relatively low. So to apply the use of biopolymers in biomedical devices, electronics, packaging, the food industry and many more, it is important to incorporate reinforcement materials, which will greatly improve the properties of these environmentally friendly composites [16]. These products are biocompatible, non-toxic, biodegradable and easily recyclable, which allows us to label them as sustainable [17]. All listed features make biopolymers a very promising solution for biomedical applications. Biopolymers can be classified as natural or synthetic based on their origin. The most commonly used ones are polylactic acid (PLA) and polyhydroxyalkanoate (PHA), which are widely studied and tested [18], but many others are also very promising but have not yet been well researched. Methods of application of biopolymers are generally categorized as bioprinting and include such techniques as fused deposition modelling (FDM), selective laser sintering (SLS), computer-aided wet spinning (CAWS), stereolithography (SLA) and spin coating [19]. By now, biopolymers have been used in such medical applications as soft-tissue replacement vascular grafts, breast implants, intraocular lenses, artificial hearts, dialyzers, catheters, external and internal ear repairs, coatings for pharmaceutical tablets and capsules, cardiac assist devices, implantable pumps, pacemakers, heart valves, drug delivery, and targeting sites of tumours or inflammation [20]. But there are surely many more applications soon to be established.

Taking into account all the discussed aspects, any additional knowledge and techniques in the field of biopolymers with the incorporation of reinforcement materials in printed electronics are desirable. For the aim of this work, we conducted research on the possibilities of using a polyhydroxybutyrate/polyhydroxyvalerate (PHB/PHV) biopolymer with the addition of graphene nanoplatelets in printed electronics technology. This polymer is a copolymer of two thermoplastic materials obtained by biological fermentation from renewable carbohydrate raw materials. The PHB homopolymer is a rigid and brittle

polymer with high crystallinity, whose mechanical properties resemble those of polystyrene, although it is less brittle and more temperature-resistant. Polyhydroxyvalerate (PHV) units act to lower the melting point, increase impact strength and flexibility, but also reduce the tensile strength. PHB/PHV copolymers are used instead of PHB homopolymers to achieve a better balance between stiffness and strength. PHV contents of 5%–20% give properties generally similar to those of polyolefins. They melt at lower temperatures than homopolymer, giving a useful improvement in melt processability. They are used for biodegradable containers (of which shampoo bottles are the most common example) and other items that are difficult to recycle, such as disposable razors or medically contaminated items. It is believed that the medical/biological fields provide the most potential application areas for this polymer [21–23].

There are no reports of the use of this biopolymer in applications for conductive layers applied by printed electronics technologies. Given its biodegradability and mechanical properties, it shows great potential for use as a matrix for conductive layers containing a functional carbon phase, fitting into the global trend of sustainable electronics. Researchers suggest that the electrical conductivity of PHB/PHV biopolymer can be enhanced by combining it with carbon nanoparticles. However, this has not been tested yet [24].

In this work, we have investigated the feasibility of producing conductive pathways based on PHB/PHV and graphene nanoplatelets through a spray-coating method. We tested various solvents, different functional phase concentrations and the influence of deposition temperature on the layer properties. The tracks were applied to both a PET substrate and an environmentally friendly paper substrate and then tested for their electrical properties and adhesion to the substrate.

## 2. Materials and Methods

### 2.1. Materials

Biopolymer polyhydroxybutyrate/polyhydroxyvalerate 2%, which, according to information obtained from the manufacturer, means 2 wt% PHV content, with the chemical structure as shown in Figure 1, in the form of powder with a maximum particle size of 300 μm and molecular weight of 410 kg/mol, was purchased from Goodfellow (Incheon, Republic of Korea). Graphene nanoplatelets with thicknesses of 8–15 nm and diameters of 1–2 μm were obtained from Cheap Tubes Inc. (Cambridgeport, VT, USA). Dichloromethane (DCM) with a density of 1.33 $g/cm^3$ and boiling point at 40 °C, Trichloromethane/Chloroform D1 with TMS with a density of 1.5 $g/cm^3$ and boiling point at 61.5 °C and N,N-Dimethylformamide (DMF) with a density of 0.95 $g/cm^3$ and boiling point at 153 °C were purchased from Carl Roth GmbH + Co. KG (Karlsruhe, Germany). Surfactant MALIALIM® AKM-0531 with cytocompatibility confirmed by research was obtained from NOF EUROPE GmbH (Frankfurt am Main, Germany). Silver conductive paste LOCTITE® EDAG PM 460A E&C was purchased from Henkel Adhesives (Düsseldorf, Germany). Mylar® PET foil with a thickness of 35 μm was purchased from Dupont Teijin Films (Dumfries, UK). As a second substrate, regular printing paper with a grammage of 120 $g/m^2$ was used.

**Figure 1.** Chemical structure of Polyhydroxybutyrate/polyhydroxyvalerate (PHB/PHV) biopolymer [24].

## 2.2. Methods

### 2.2.1. Preparation of the Stencil

The stencil pattern was prepared in the Inkscape program and then cut from a 2 mm-thin PMMA sheet using a laser engraving machine with a 30 W $CO_2$ laser. The stencil template included paths in two lengths of 25 mm and 50 mm and three widths of 1 mm, 2 mm and 5 mm, for five of each.

### 2.2.2. Fabrication of the Biopolymer Solution

Six samples of the solution were prepared with three different solvents. The examined percentages of biopolymer in the solution were 1% and 2.5%. First, the necessary amount of biopolymer was weighed into the container to which the solvents were added. The container was secured against evaporation of the solvent, and then the solution was stirred with a magnetic stirrer for at least 24 h at a temperature not exceeding the boiling point of each of the solvents.

### 2.2.3. Fabrication of the Graphene Nanoplatelets Suspension

The examined percentage of graphene nanoplatelets in the solution was 0.5%, 1% and 1.5%, and these exact amounts were weighed into a container separate from the biopolymer solution. It was followed by the addition of each of the three solvents and a surfactant to obtain the appropriate degree of deagglomeration of graphene nanoplatelets. Then, the solution was sonicated in a homogenizer for 10 min with an amplitude of 90%. Next, suspensions were prepared with the same procedure with two chosen solvents (dichloromethane and chloroform) and three different concentrations of graphene nanoplatelets: 0.5%, 1% and 1.5%. Samples with chloroform as solvent were additionally heated up to 60 °C, whereas dichloromethane was not heated due to its relatively low boiling point.

### 2.2.4. Preparation of the Biopolymer-Graphene Composite

After preparing the biopolymer solution and graphene nanoplatelets suspension separately, samples with corresponding solvents were placed together in one beaker. The composition was sonicated in a homogenizer for 4 min with 90% amplitude. Immediately after preparation, the composite had to be spray-coated through a suitable mask onto substrates, because otherwise the phases would separate and the solution would have to be mixed again.

### 2.2.5. Application of Composite Layers

The composite layers were fabricated using the spray-coating method with an airbrush with a pressure of 0.6 MPa at a specially prepared laboratory station. The conductive tracks were applied with a uniform motion with the nozzle fully open (fully retracted needle). Because of the liquid splashing at the start of the spraying process, each time the jet was positioned just outside of the stencil and then directed onto the pattern, thereby obtaining a uniform layer. To measure the electrical properties of the layers, silver connectors were applied using a thermoplastic, rapid-drying, flexible and conductive coating.

### 2.2.6. Characterization

Electrical measurements were carried out using a digital meter with a range of up to 40 MΩ. As an unambiguous measurement of electrical parameters, the sheet resistance expressed in Ohms per square (Ω/□) was calculated according to the formula:

$$R_s \left[ \frac{\Omega}{\Box} \right] = R \cdot \frac{W}{L}, \tag{1}$$

where:
$R_s$—sheet resistance,
$R$—measured resistance,

*W*—width of the measured track,
*L*—length of the measured track.

Layer thickness and surface topography were evaluated with a Bruker (Billerica, MA, USA) DektakXT stylus contact profilometer using a stylus with a tip radius of 12.5 μm. The stylus force was set to 5 mg, which corresponds to $4.9 \times 10^{-5}$ N. The horizontal resolution along the scanning direction was 0.11 μm/pt, and the vertical scanning range of the device was set at 65.5 mm.

Scan locations and profile analysis were performed using a semi-automatic procedure to ensure comparability of results. Three scans each were taken at an interval of 1.5 μm symmetrically to the path centre. Ten samples were examined for each concentration of graphene.

After scanning, the roughness and layer height were calculated. Segments of the unprinted substrate on both sides of the path were used to align the profile characteristics and set a zero point for height estimation. The segment of the printed layer was used to calculate the average height (layer thickness) and arithmetic average roughness (Ra) (Figure 2).

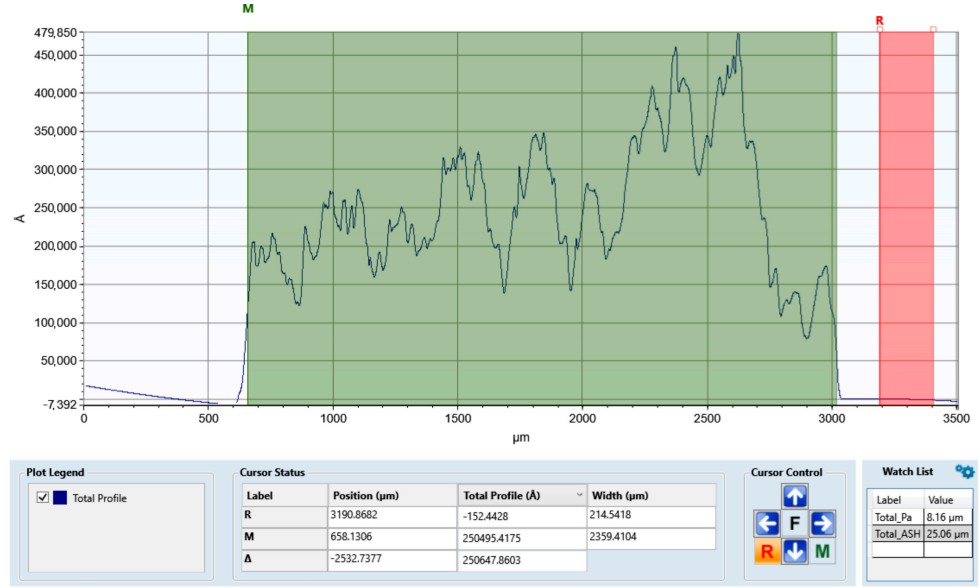

**Figure 2.** A representative scan for a print on a PET substrate with a layer sprayed with a composite containing 1.5% GNP based on heated chloroform. The green colour indicates the measurement range, that is, the path area, the red colour indicates the reference, that is, the substrate area.

Electrical conductivity, expressed in Siemens per meter (S/m) was calculated according to the formula:

$$\sigma \left[ \frac{\text{S}}{\text{m}} \right] = \frac{L}{R \cdot A} = \frac{L}{R \cdot W \cdot T} = \frac{1}{R_s \cdot T}, \tag{2}$$

where:
$\sigma$—electrical conductivity,
*R*—measured resistance,
*A*—cross-sectional area of the measured track,
*W*—width of the measured track,
*L*—length of the measured track,
$R_s$—sheet resistance,
*T*—layer thickness.

Adhesion measurements were carried out according to the ISO 2409 norm using a 2 mm coating knife, which provided the blades to be at a constant angle of contact with the surface. The ISO 2409 standard is a destructive system for evaluating the adhesion of a

layer to a surface. It consists of 6 levels of adhesion in the range from 0 to 5, where 0 means no peeling from the surface and 5 corresponds to no adhesion, meaning more than 60% of the layer has been peeled off.

Optical density and CIE *L*a*b*\* values were measured using an X-Rite eXact spectrophotometer under the following conditions: D50 luminant, 2° colourimetric observer and M2 (UVC) filter. The white ink-jet paper was used as the background for the measurement of layers on foil substrates. The measurement was repeated 5 times.

Scanning electron microscopy (SEM) was conducted on a Hitachi SU8230 (Hitachi High-Tech Europe GmbH, Krefeld, Germany) instrument with an accelerating voltage of 7.0 kV and an upper secondary electron detector.

## 3. Results and Discussion

### 3.1. Sheet Resistance Results

Given that the main goal of the presented work was to evaluate the potential use of the polyhydroxybutyrate/polyhydroxyvalerate 2% biopolymer in printed electronics, efforts were made to discover a proper composition and concentrations of components that ensure good electrical conductivity. For this purpose, a series of samples with different solvents, modifying agents and different concentrations of both biopolymer and graphene nanoplatelets were prepared.

The first tests were carried out for two concentrations of biopolymer in the solutions of 1% and 2.5% and for the 0.5% concentration of graphene nanoplatelets. Samples were applied onto two substrates: PET foil and paper. During the application of DMF solvent-based solutions, runoff of the composite from the substrate was observed for both paper and film, i.e., failure to maintain the pattern imposed by the template (Figure 3b). This was most likely caused by the high evaporation temperature of this solvent, too high for use in the spray-coating method. Since the dissolved film is not suitable for measurements, the decision was made to eliminate DMF-based solutions from subsequent testing. Almost every measurement exceeded the measuring range of the used digital meter, meaning that printed layers of these solutions have a resistance over 40 MΩ, resulting in negligible or even non-existent electrical conductivity. Only the solution composed of 1% biopolymer, 0.5% graphene nanoplatelets and chloroform as a solvent showed a resistance of 117 kΩ/□ on paper and a resistance of 237 kΩ/□ on PET foil. The very high sheet resistance obtained may result from a low layer adhesion to the substrate. Resistance on paper is nearly two times smaller than on foil, which may be the consequence of the greater porosity of the paper, leading to improved adhesion of the composite.

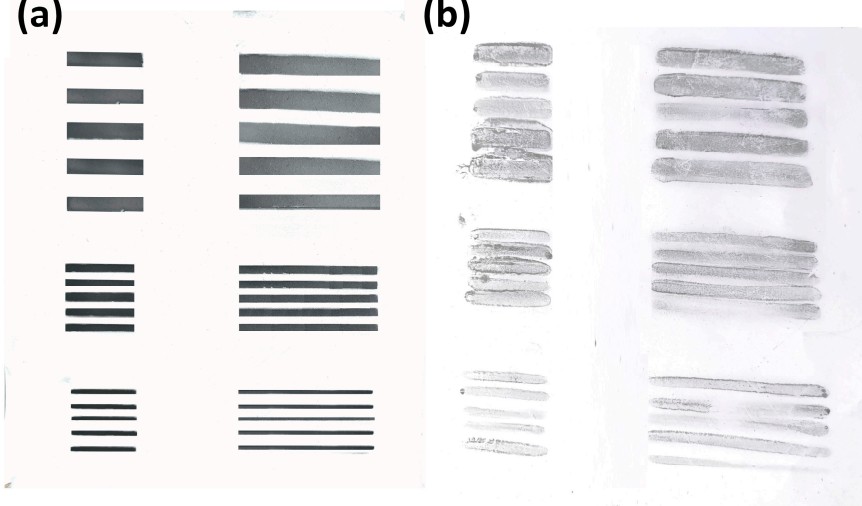

**Figure 3.** Picture illustrating the proper imprinting (**a**) of the test pattern when chloroform was used and the improper printing (**b**) obtained when DMF was used.

The last of the conducted tests was to establish the influence of graphene nanoplatelet concentration on the electrical properties of the solution. Based on previously carried out experiments, only one of the tested concentrations of polyhydroxybutyrate/polyhydroxyvalerate 2% was chosen. Solutions for this series of samples were prepared with 1% biopolymer, 0.5%, 1% and 1.5% concentrations of graphene nanoplatelets and DCM and chloroform as solvents. Solutions were spray-coated on two substrates: paper and PET foil. Additionally, to increase the solubility of the biopolymer, the samples with chloroform were heated to 60 °C. For samples with DCM, heating was not possible due to the low boiling point of the solvent. The samples with 0.5% graphene and DCM as a solvent showed high sheet resistance, exceeding 40 MΩ. The rest of the results are presented in Table 1.

**Table 1.** Sheet resistance of 1% biopolymer composition with different concentrations of graphene.

| Graphene (%) | Solvent | Substrate | Resistance (kΩ/□) |
|---|---|---|---|
| 0.5 | Chloroform 60 °C | paper | 2.65 |
| | | foil | 4.59 |
| 1 | DCM | paper | 8.02 |
| | | foil | 3.98 |
| | Chloroform 60 °C | paper | 0.27 |
| | | foil | 0.25 |
| 1.5 | DCM | paper | 1.64 |
| | | foil | 2.11 |
| | Chloroform 60 °C | paper | 0.099 |
| | | foil | 0.096 |

The results in Table 1 show that, as the concentration of graphene nanoplatelets in the composition increases, a reduction in sheet resistance occurs. This is an expected result, as increasing the proportion of the electrically conductive phase enhances electron transport in the layer. This trend persists for both solvents used; however, the values obtained for heated chloroform are significantly higher than for DCM. The heating process increases the solubility of the biopolymer, resulting in better adhesion and better electrical connections between graphene flakes in the spray-coated pathways. For 1% and 1.5% graphene concentrations, in almost every case the resistance is smaller on film than on paper, the only exception being 1.5% graphene in DCM solvent. This is most likely a consequence of the absorption properties of the paper substrate. As the layer soaks into the paper substrate, it covers a wider volume, thereby distancing the individual particles of the functional phase and impeding current flow.

Heating chloroform in the last series of tests allowed us to examine the effect of the composite temperature on the electrical properties of the printed layers. For this purpose, the results of resistance measurements from the first phase of tests were compared with the corresponding heated composites in the last phase, which are presented in Table 2. The compared composites had a 1% concentration of biopolymer and 0.5% concentration of graphene.

**Table 2.** Sheet resistance of 1% biopolymer/0.5% graphene composition with different temperatures of solvent.

| Substrate | Chloroform Temperature | Resistance (kΩ/□) |
|---|---|---|
| paper | room | 117 |
| | 60 °C | 2.65 |
| foil | room | 237 |
| | 60 °C | 4.59 |

The comparison presented in Table 2 shows that the heating process decreased the sheet-resistance values of composition for both paper and foil by a factor of about fifty.

This might arise from the fact that the heated solvent evaporated faster than in room-temperature deposition, causing a larger amount of polymer and graphene to be deposited over the same time.

### 3.2. Adhesion Results

Observing the significant differences in sheet resistance results with a fixed functional phase and polymer content and changed vehicle solvent, along with changes in application temperature within one solvent, a conclusion can be drawn that both the solubility of the polymer in the solvent and the rate of its evaporation during the spraying process are crucial factors. However, on the other hand, the adhesion of the film to the substrate can play a major role as well. To verify adhesion, tests were performed for layers containing different graphene contents and applied with both DCM and heated chloroform as solvents. Adhesion measurements were conducted according to the ISO 2409 norm, and the results are in Table 3.

**Table 3.** Adhesion measurement results for composition with 1% of biopolymer.

| Graphene [%] | Solvent | Substrate | Adhesion ISO 2409 |
|:---:|:---:|:---:|:---:|
| 0.5 | DCM | paper | 5 |
|  |  | foil | 5 |
|  | Chloroform 60 °C | paper | 3 |
|  |  | foil | 2 |
| 1 | DCM | paper | 5 |
|  |  | foil | 5 |
|  | Chloroform 60 °C | paper | 3 |
|  |  | foil | 2 |
| 1.5 | DCM | paper | 5 |
|  |  | foil | 5 |
|  | Chloroform 60 °C | paper | 3 |
|  |  | foil | 2 |

The results presented in Table 3 show that increasing the concentration of graphene nanoplatelets has no effect on the layer adhesion to the substrate. Measurements for the paper substrate were complicated by the fact that the substrate layer was detached along with the removal of the scotch tape necessary in the adhesion measurement procedure inside the chosen method. However, there is a clearly visible difference for both paper and film substrates under different solvents tested. Such differences in adhesion may arise from the improved dissolution of the tested biopolymer in the heated chloroform, resulting in an increase in adhesion to the substrate. With such results, the use of heated chloroform as a solvent for the polyhydroxybutyrate/polyhydroxyvalerate polymer in the preparation of conductive paths based on graphene nanoplatelets seems to be accurate.

### 3.3. Layer Thickness and Electrical Conductivity Measurements

Measurements of film thickness are often omitted in studies, yet this is a highly important parameter which is used to calculate such parameters as electrical conductivity, among others. Depending on the type of media used and the content of the functional phase, the thickness may vary significantly. Therefore, it is of great importance to measure the thickness of the tested paths and take it into account when measuring electrical properties.

The thickness of the pathways was studied for all three concentrations of graphene nanoplatelets with both heated chloroform and DCM as solvents on film substrates (Table 4). Measurements on paper substrates were unreliable due to their high surface roughness.

**Table 4.** Conductivity and layer thickness of 1% biopolymer composition with different concentrations of graphene with chloroform 60 °C as a solvent and foil as substrate.

| Graphene (%) | Layer Thickness (µm) | Conductivity (S/m) |
|---|---|---|
| 0.5 | 8.79 | 24.80 |
| 1 | 15.00 | 266.76 |
| 1.5 | 25.22 | 413.01 |

The results in Table 4 show a direct correlation between the packing of graphene nanoplatelets and the measured path thickness. As the percentage of the functional phase increases, the thickness of the measured path increases. No differences were noticed in the obtained thickness depending on the solvent used. The film thickness result obtained for 1.5% graphene concentration (~25 µm) is commonly used as a standard film thickness for sheet-resistance measurements.

In the scientific literature, one can find reports of electrical conductivities for graphene-containing inks of approximately $10^4$ S/m; however, such values were obtained using a matrix of highly conductive polymers [25]. The results obtained by Carey Tian et al. show that for graphene films that do not contain conductive polymers, the conductivities of spray-coated films are about 3 S/m [26].

In our study, the best layer resistivity values were characterized by the layers applied for the post-heated chloroform. The electrical conductivity was calculated for these samples (Table 4). The best conductivity was obtained for 1.5% graphene content and amounted to 413 S/m, which is as expected, as graphene nanoplatelets constitute an electrically conductive functional phase in the layer. However, even for 0.5%, the conductivity is almost seven times higher than in the competitive research mentioned above.

*3.4. Colour Parameters and Optical Density Measurements*

The colour parameters and optical density of measured lines are summarized in Table 5. These parameters are useful for assessing the quality of the print and can be transferred to determine the quality of conductive tracks. The colour values *L\**, *a\**, *b\**, and optical density allow us to assess the colour and the thickness of track layers, respectively. The *L\** value is responsible for lightness (from black 0 to white 100); *a\** (from green −128 to red 127), and *b\** (from blue −128 to yellow 127).

**Table 5.** Colour parameters *L\**, *a\**, *b\**, optical densities of tracks.

| Graphene [%] | Solvent | Substrate | *L\** | *a\** | *b\** | ΔE [1] | Optical Density |
|---|---|---|---|---|---|---|---|
| 0.5 | Chloroform 60 °C | paper | 36.36 | −0.15 | −1.03 | 5 | 0.91 |
| | | foil | 41.48 | −0.05 | −1.04 | | 0.91 |
| 1 | Chloroform 60 °C | paper | 41.73 | 0.29 | −0.12 | 1 | 0.91 |
| | | foil | 41.58 | −0.08 | −1.00 | | 0.91 |
| 1.5 | Chloroform 60 °C | paper | 38.58 | 0.43 | 0.15 | 2 | 0.99 |
| | | foil | 36.36 | 0.62 | 0.55 | | 1.04 |

[1] Colour differences ΔE were calculated from $\Delta E = \sqrt{(\Delta L^*)^2 + (\Delta a^*)^2 + (\Delta b^*)^2}$, where $\Delta L^*$, $\Delta a^*$, $\Delta b^*$ are the differences between the colour values of the layer on the paper and foil.

The optical density is in the range of 0.91 to 1.04. Slightly higher values are observed for the ink with 1.5% graphene, 0.99 and 1.04, on paper and foil substrates, respectively. This is related to a slightly thicker layer of the coating.

The increase of the graphene content from 0.5% to 1.5% in the ink layer gives a slight difference in the *b\** parameter, which changes from −1.04 for the foil substrate up to 0.55, which indicates the change of the colour from light blue to a more yellowish colour. Colour differences ΔE were calculated as the difference between the colour of the layer on the

paper and foil substrate. When ΔE is higher than 1, the change of the layer colour can be noticeable to the naked human eye.

Conclusively, it should be stated that the conducted tests prove a good packing of the layer with graphene nanoplatelets for all tested concentrations and both substrates. Each of the tested samples showed good electrical and adhesion properties.

### 3.5. SEM Imaging

Scanning electron microscope imaging was performed to further complement the optical density test with observations of the layer structure. For chloroform-based samples heated to 60 °C, SEM imaging was performed for the three concentrations of graphene nanoplatelets tested (Figure 4).

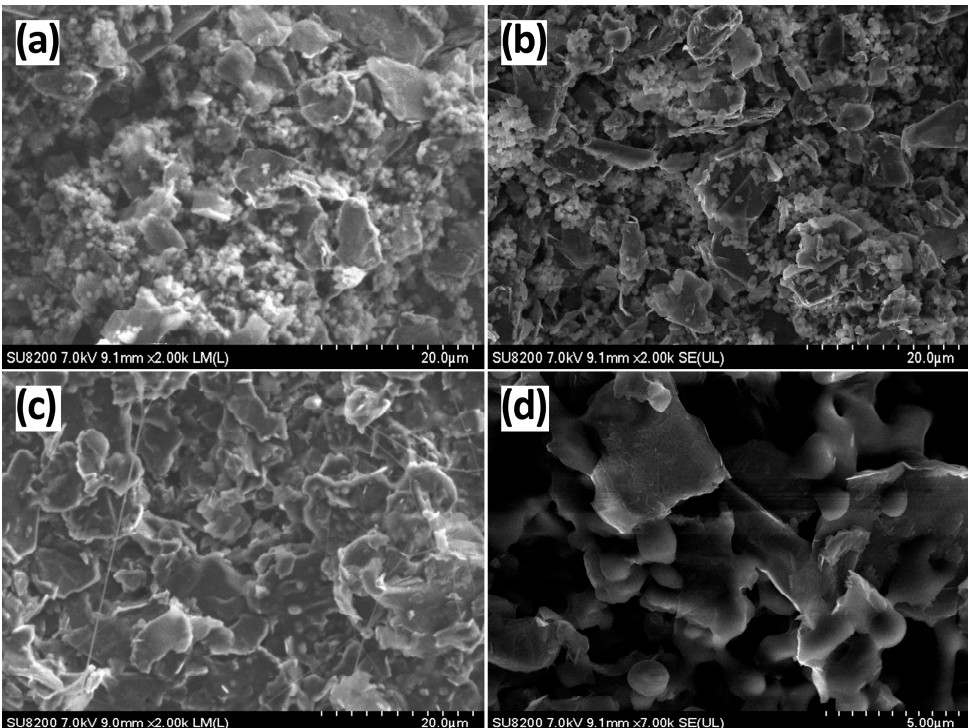

**Figure 4.** SEM images of layers deposited with a 1% PHB/PHV suspension in chloroform, heated to 60 °C for (**a**) 0.5%; (**b**) 1%; (**c**,**d**) 1.5% graphene nanoplatelet content.

The SEM images clearly show a homogeneous distribution of graphene flakes relative to the polymer molecules. As the percentage of graphene content increases, the flakes dominate more in the final layer. This demonstrates the correctness of the assumptions and application of the spray-coating method to the tested graphene flake suspension in a 1% PHB/PHV solution in chloroform, as it indicates the lack of nozzle clogging by graphene flakes. It is also consistent with the sheet-resistance measurements.

It can also be seen that, for each concentration of graphene flakes, the packing of the flakes within the layer is high and enables the flow of charge within the layer. In Figure 4d, for higher magnification, the contact between the individual graphene flakes with each other is visible.

### 4. Conclusions

This work aimed to identify the possibility of using the polyhydroxybutyrate/polyhydroxyvalerate 2% biopolymer in printed electronics. For this purpose, tests determining electrical properties and adhesion were conducted on samples applied by the spray-coating method.

As materials for this research, three solvents were chosen: Dichloromethane (DCM), Chloroform and N,N-Dimethylformamide (DMF); different concentrations of both biopolymer and graphene nanoplatelets were also included. Conducted tests show that the N,N-Dimethylformamide (DMF) solvent is not suitable for the spray-coating method. It may be due to its high boiling point, which is unfavourable for this printing technique, causing the composition to flow on the substrate instead of forming a uniform layer.

Based on testing of different-percentage packing of the functional phase, the conclusion may be drawn that all tested concentrations of graphene nanoplatelets produce an electrically conductive layer with good optical density, indicating good coverage of the substrate. However, the higher the concentration of the functional phase, the higher the thickness of the layer and the greater its electrical conductivity.

The studies outlined above revealed that the best solvent for obtaining electrically conductive paths based on polyhydroxybutyrate/polyhydroxyvalerate 2% is chloroform heated to 60 °C. The best electrical performance was obtained for this system.

The best conductivity was achieved with 1.5% graphene, reaching 413 S/m. However, even at 0.5%, the conductivity is almost seven times higher than in the competitive research [25,26].

The tests conducted on colour parameters and optical density prove a good packing of the layer with graphene nanoplatelets for all tested concentrations and both substrates. Each of the tested samples showed good electrical and adhesion properties.

Taking all measured parameters into consideration, the most promising layers were made using a 1% concentration of the polyhydroxybutyrate/polyhydroxyvalerate 2% biopolymer, 1.5% of graphene nanoplatelets and chloroform heated to 60 °C as a solvent. The solution was printed on a foil substrate using the spray-coating method and shows both good electrical properties and adhesion. The conducted research opens a wide path for the use of polyhydroxybutyrate/polyhydroxyvalerate 2% biopolymer supplemented with graphene nanoplatelets in biomedical applications, ensuring good conductivity, biocompatibility and stability.

**Author Contributions:** S.L.-K.: Conceptualization, methodology, formal analysis, writing—review and editing, K.W.: investigation, data curation, writing—original draft preparation; D.B.: investigation, data curation; A.K.: investigation, data curation; T.T.M.: investigation, data curation; D.J.: investigation, data curation; M.J.: supervision, funding acquisition. All authors have read and agreed to the published version of the manuscript.

**Funding:** The research was supported by EMERGE project, funded from the European Union's Horizon 2020 research and innovation programme under grant agreement No. 101008701.

**Institutional Review Board Statement:** Not applicable.

**Informed Consent Statement:** Not applicable.

**Data Availability Statement:** Data available upon request.

**Conflicts of Interest:** The authors declare no conflict of interest.

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
