# Peer review of "Biodegradable Conductive Layers Based on a Biopolymer Polyhydroxybutyrate/Polyhydroxyvalerate and Graphene Nanoplatelets Deposited by Spray-Coating Technique"

_coatings, doi:10.3390/coatings13101791_

Round 1
Reviewer 1 Report
The paper is descriptive and results and discussions are well correlated. However for better clarity and wider readership specific comments must answered from authors side and edit too:
1. What is the novelty and main question addressed by the research?
2. Abstract should be specific topic oriented. Do you consider the topic original or relevant in the field?
3. Comparative data or Table must supplied to address a specific gap in the field?
4. Add relevant references published recently in this subject area with clearcut comparison?
5. Draw special attention towards research methodology to improve quality of papers.
6.Polished whole manuscript particularly captions (figure/Table)
7.Are the conclusions consistent with the evidence and arguments presented and do they address the main question posed?
8. Figure quality needs to be improve
9. Conclusion portion needed widerness and may correlated with current theme.
Author Response
We would like to thank the reviewer for their insightful and helpful comments, which are reproduced below in italics.
Responses to the comments of Reviewer 1
Comment 1:
What is the novelty and main question addressed by the research?
Response to comment 1:
The main goal of the research was to establish the potential of a composite based on a Polyhydroxybutyrate/Polyhydroxyvalerate biopolymer and graphene nanoplatelets for usage in printed electrically conductive structures. The novelty of the presented results is manifested by the lack of research concerning this particular biopolymer in the field of printed electronics and by the biodegradability of all materials used (biopolymer, graphene and paper substrate). That implies that conductive structures produced with tested materials have a huge potential for biodegradability, hence for being environmentally friendly. When implemented in biomedical devices, our solution can contribute to reducing large amounts of waste.
Comment 2:
Abstract should be specific topic oriented. Do you consider the topic original or relevant in the field?
Response to comment 2:
The abstract has been rewritten as follows (page 1 of the revised manuscript).
“In light of the growing concern for environmental protection and the alarming amount of waste produced due to hygiene regulations, this study suggests a biodegradable and eco-friendly solution that could make a significant contribution to the preservation of our planet. The developed solution was based on the Polyhydroxybutyrate/Polyhydroxyvalerate biopolymer, which has been tested regarding its physicochemical parameters and possible use in printed electrically conductive structures. Graphene nanoplatelets have been used as the conductive functional phase, due to literature reports of their potential use in biomedical applications and due to the potential of providing cytocompatibility in electrical structures by carbon nanomaterials. Prepared composite have been spray-coated onto PET film and paper substrates, and then subjected to electrical, adhesion and optical measurements. In order to establish the conductivity of the developed composite its resistance, layer thickness and surface topography were measured. Optical parameters have been specified using scanning electron microscopy (SEM) imaging and spectrophotometry. Conducted research opens a wide path for the use of Polyhydroxybutyrate/Polyhydroxyvalerate biopolymer with graphene nanoplatelets in biomedical applications, ensuring good conductivity, biocompatibility and stability.”
We believe that the presented topic is both original and relevant in the field because studies do not show the usage of PHB/PHV biopolymer in printed electrically conductive structures. As shown in the results of our work, the developed solution has very promising properties and after further evaluation, it can be widely used in the field of biomedical engineering. Our research shows other scientists that this particular biopolymer, after proper functionalisation, can be an eco-friendly, biodegradable and biocompatible solution.
Comment 3:
Comparative data or Table must supplied to address a specific gap in the field?
Response to comment 3:
During the literary discernment, we have not encountered any reports of the usage of PHB/PHV biopolymer in the field of printed electronics for printing electrically conductive structures. We found and compared to our work the electrical properties of spray-coated structures produced with graphene nanoplatelets and different non-conductive polymers. Relevant references for this particular comparison have been implemented in the section “3.3. Layer thickness and electrical conductivity measurements” on page 9 of the revised manuscript.
Comment 4:
Add relevant references published recently in this subject area with clearcut comparison?
Response to comment 4:
As mentioned in the response to comment 3, there are no scientific publications on the usage of PHB/PHV biopolymer in the field of printed electronics. We have included publications regarding the used method of printing – spray-coating with graphene nanoplatelets:
Carey, T.; Jones, C.; Le Moal, F.; Deganello, D.; Torrisi, F. Spray-Coating Thin Films on Three-Dimensional Surfaces for a Semitransparent Capacitive-Touch Device. ACS Appl. Mater. Interfaces 2018, 10, 19948–19956, doi:10.1021/acsami.8b02784.
We have also added a publication that states the possibility of combining PHB/PHV biopolymer with carbon nanoparticles, which may result in enhancing its non-existent electrical conductivity:
Rivera-Briso, A.L.; Serrano-Aroca, Á. Poly(3-Hydroxybutyrate-Co-3-Hydroxyvalerate): Enhancement Strategies for Advanced Applications. Polymers (Basel) 2018, 10, 732, doi:10.3390/polym10070732.
Comment 5:
Draw special attention towards research methodology to improve quality of papers.
Response to comment 5:
All methods implemented during the research are considered to be standard in the field of printed electronics. Tests have also been carried out according to current standards. Research methodology has been thoroughly discussed in the section “2.2. Methods”. If any more specific information is needed we will be pleased to complement this section.
Comment 6:
Polished whole manuscript particularly captions (figure/Table)
Response to comment 6:
The whole manuscript was thoroughly revised for any misspellings or formatting issues. In the revised manuscript all found mistakes have been corrected and we made sure the formatting was consistent with the given regulations.
Comment 7:
Are the conclusions consistent with the evidence and arguments presented and do they address the main question posed?
Response to comment 7:
All conclusions were reached by analysing the results of the conducted research. The main question was whether PHB/PHV biopolymer can be used in the field of printed electronics. Drawn conclusions show that specific parameters can ensure good conductivity, biocompatibility and stability of solution. So in our opinion, it is safe to say that drawn conclusions address and confirm the question posed in the publication.
Comment 8:
Figure quality needs to be improve
Response to comment 8:
All included figures have been re-uploaded to the manuscript after verification and corrections. We made sure all pictures’ resolutions met the standards set by the journal.
Comment 9:
Conclusion portion needed widerness and may correlated with current theme.
Response to comment 9:
Thank You for this insightful comment. We revised the stated conclusions, applied corrections and added more details regarding all results in the revised manuscript (page 11). Drawn conclusions address and confirm the main question we stated in this research. We have demonstrated that it is possible to create an electrically conductive printed structure using PHB/PHV biopolymer with appropriate parameters and functionalization. We have demonstrated that by adjusting the parameters and introducing additional functionalization, it is possible to produce an electrically conductive printed structure using PHB/PHV biopolymer.
Reviewer 2 Report
In this work, the authors reported the biopolymer PHB/PHV based composite to serve as the biodegradable conductive layer. However, the content is not consistent with the title and mislead the readers. The work lack of significant novelty required by the journal and not recommend to publish at current stage.
1. The polymer chemical structures of PHB/PHV should be provided.
2. The biodegradability of the polymer composite should be studied.
Minor editing of English language required.
Author Response
We would like to thank the reviewer for their insightful and helpful comments, which are reproduced below in italics.
Responses to the comments of Reviewer 2
Comment 1:
The polymer chemical structures of PHB/PHV should be provided.
Response to comment 1:
We complemented the “2.1. Materials” section in the revised manuscript with the chemical structure of PHB/PHV biopolymer. (page 3)
Comment 2:
The biodegradability of the polymer composite should be studied.
Response to comment 2:
Thank You for noting this important aspect. Further research in the future may include studies of the biodegradability of this polymer composite. But for now, our goal was to establish the potential of a composite based on a PHB/PHV biopolymer and graphene nanoplatelets for usage in printed electrically conductive structures. Hence the electrical properties were our main focus. Choosing a biopolymer supplemented with graphene nanoplatelets (which are carbon nanoparticles with 97% purity) and paper as a substrate leads to the conclusion that a solution produced with these ingredients has great potential for biodegradability.
Reviewer 3 Report
In this work, the authors reported a conductive composite of Poly- 2 hydroxybutyrate/Polyhydroxyvalerate and graphene nano-platelets deposited by spray coating technique. However, the data is not sufficient to establish the conductivity of composites.very few samples are prepared. The combination of Poly- 2 hydroxybutyrate/Polyhydroxyvalerate is not justified. Thermal properties and biodegradability are not deported. Hence the article is not suitable to be published in “coatings”
Author Response
Dear Reviewer. The manuscript was revised after comments from three other reviewers. We hope that the changes made will convince you about the research and the reasonableness of its publication in "Coatings"
Reviewer 4 Report
The goal of the paper is to study the potential of a composite based on a biodegradable copolymer (poly(hydroxybutyrate-hydroxyvalerate)) and graphene nanoplatets to be used in printed electrically conductive structures. The authors study the sheet resistance, adhesion and electrical conductivity colour parameters and optical density. Different solvents, percentage of polymer and graphene have been studied.
The subject is of interest due to the degradability and biobased origin of the copolymer. However, in order to be published it needs minor revision. Some remarks are given in following lines:
- The abstract has to be rewritten because it contains too many introductory lines, and instead, more description of the results obtained should be included.
- Line 122 The polymer used is “Polyhydroxybutyrate/Polyhydroxyvalerate 2%”. What does 2% stand for? PHV w%, mol%... ? This should be clarified.
- Line 228. It is explained that different modifying agents has been used, however only one type of surfactant has been used. This should be modified.
- Line 354. The sentence that starts in line 354 and finishes in line 356 has to be removed because it is duplicated.
- Line 366. The sentence that starts in line 366 and finishes in line 368Line has to be removed because it is duplicated.
- Line 388. Instead of X(d) it should be written 3(d).
Author Response
We would like to thank the reviewer for their insightful and helpful comments, which are reproduced below in italics.
Responses to the comments of Reviewer 4
Comment 1:
The abstract has to be rewritten because it contains too many introductory lines, and instead, more description of the results obtained should be included.
Response to comment 1:
The abstract has been rewritten in the revised manuscript. The content of the corrected paragraph was included in the answer to Reviewer 1's second question.
Comment 2:
Line 122 The polymer used is “Polyhydroxybutyrate/Polyhydroxyvalerate 2%”. What does 2% stand for? PHV w%, mol%... ? This should be clarified.
Response to comment 2:
According to information obtained from the manufacturer, 2% means the weight percentage of PHV content. We have added this information in the manuscript
Comment 3:
Line 228. It is explained that different modifying agents has been used, however only one type of surfactant has been used. This should be modified.
Response to comment 3:
Thank You for noticing. This mistake has been modified in the revised manuscript (page 6, line 228).
Comment 4:
Line 354. The sentence that starts in line 354 and finishes in line 356 has to be removed because it is duplicated.
Response to comment 4:
Thank You for noticing. The duplicated sentence has been removed from the revised manuscript (page 9).
Comment 5:
Line 366. The sentence that starts in line 366 and finishes in line 368Line has to be removed because it is duplicated.
Response to comment 5:
Thank You for noticing. Duplicated sentences have been removed from the revised manuscript (page 9).
Comment 6:
Line 388. Instead of X(d) it should be written 3(d).
Response to comment 6:
Thank You for noticing. This correction has been applied in the revised manuscript (page 10).
Round 2
Reviewer 2 Report
The manuscript can be accepted
Reviewer 3 Report
the authors reviewed the manuscript with the available data. the article may be accepted
Reviewer 4 Report
I consider that the reviewers remarks have been correctly adressed and the paper can be published as it is.